# Development of PET Radioisotope Copper-64-Labeled Theranostic Immunoliposomes for EGFR Overexpressing Cancer-Targeted Therapy and Imaging

**DOI:** 10.3390/ijms25031813

**Published:** 2024-02-02

**Authors:** Hwa Yeon Jeong, Seong Jae Kang, Min Woo Kim, In-ho Jeong, Moon Jung Choi, Cheulhee Jung, In Ho Song, Tae Sup Lee, Yong Serk Park

**Affiliations:** 1Department of Biomedical Laboratory Science, Yonsei University, Wonju 26493, Republic of Korea or cca1987@korea.ac.kr (H.Y.J.); sjkang332@gmail.com (S.J.K.); minwookim@yuhs.ac (M.W.K.); d4inno@naver.com (I.-h.J.); moon_jung_choi@brown.edu (M.J.C.); 2Department of Biotechnology, College of Life Sciences and Biotechnology, Korea University, Seoul 02841, Republic of Korea; damo363@korea.ac.kr; 3Division of RI-Convergence Research, Korea Institute of Radiological and Medical Science, Seoul 01812, Republic of Korea; 99269@snubh.org (I.H.S.); nobelcow@kirams.re.kr (T.S.L.)

**Keywords:** theranostics, positron emission tomography (PET) imaging, immunoliposomes, epidermal growth factor receptor (EGFR), copper-64

## Abstract

Combining standard surgical procedures with personalized chemotherapy and the continuous monitoring of cancer progression is necessary for effective NSCLC treatment. In this study, we developed liposomal nanoparticles as theranostic agents capable of simultaneous therapy for and imaging of target cancer cells. Copper-64 (^64^Cu), with a clinically practical half-life (*t*_1/2_ = 12.7 h) and decay properties, was selected as the radioisotope for molecular PET imaging. An anti-epidermal growth factor receptor (anti-EGFR) antibody was used to achieve target-specific delivery. Simultaneously, the chemotherapeutic agent doxorubicin (Dox) was encapsulated within the liposomes using a pH-gradient method. The conjugates of ^64^Cu-labeled and anti-EGFR antibody-conjugated micelles were inserted into the doxorubicin-encapsulating liposomes via a post-insertion procedure (^64^Cu-Dox-immunoliposomes). We evaluated the size and zeta-potential of the liposomes and analyzed target-specific cell binding and cytotoxicity in EGFR-positive cell lines. Then, we analyzed the specific therapeutic effect and PET imaging of the ^64^Cu-Dox-immunoliposomes with the A549 xenograft mouse model. In vivo therapeutic experiments on the mouse models demonstrated that the doxorubicin-containing ^64^Cu-immunoliposomes effectively inhibited tumor growth. Moreover, the ^64^Cu-immunoliposomes provided superior in vivo PET images of the tumors compared to the untargeted liposomes. We suggest that nanoparticles will be the potential platform for cancer treatment as a widely applicable theranostic system.

## 1. Introduction

Lung cancer is a leading cause of mortality worldwide, with non-small cell lung cancer (NSCLC) being the most prevalent and diagnosed at an advanced stage [1,2]. Platinum-based cytotoxic drugs have been the mainstay of treatment for these lung cancers. However, in advanced NSCLC, only a few drugs, such as docetaxel and pemetrexed, are used as second-line treatment options, along with epidermal growth factor receptor (EGFR) tyrosine kinase inhibitors (TKIs) like erlotinib and gefitinib [3,4]. The timely initiation of appropriate treatment is crucial for improving the survival rate. Therefore, there is a critical need to identify the most suitable drug based on treatment responsiveness.

Personalized treatment is necessary to address the unmet needs in cancer therapy. Existing treatment guidelines are based on strict standards but are derived from the average responses of numerous patients [5,6]. This approach is particularly relevant in chemotherapy, where drug overdose can have serious consequences. Hence, conducting a thorough analysis of and continuously monitoring individual patients’ drug responsiveness are imperative to tailor treatment strategies accordingly. To achieve this, a treatment platform capable of prognostication and monitoring alongside therapy is required, with theranostic nanoparticles emerging as one of the most promising platforms. Various types of nanoparticles, including peptides, polymers, and liposomes, have been actively researched as theranostic nanoparticle platforms [7,8,9].

Liposomes, which have been extensively studied for a long time, are among the most successful drug delivery systems. Liposomal doxorubicin (Doxil) was the first liposome-based drug delivery system approved by the Food and Drug Administration (FDA) [10], and liposome-based drug delivery systems continue to evolve in diverse forms. Liposomes exhibit advantageous characteristics such as in vivo applicability, ease of handling, and high stability in the body. Moreover, they can be modified into various forms, making them an attractive platform for multifunctional drug delivery systems. Traditional liposomes often utilize positively charged lipids to enhance intracellular transduction or increase drug capture efficiency [11,12]. However, these positively charged lipids can be cytotoxic and lead to non-specific binding due to the membrane charge of liposomes [13,14,15,16,17,18,19,20,21,22,23,24]. Alternatively, neutral liposomes and various drug-capture technologies have been developed [11,21,25,26,27,28]. The ongoing development of liposome-based theranostic nanoparticles highlights their potential in cancer research.

Through diverse research efforts, significant progress has been made in enhancing drug delivery efficiency and enabling real-time monitoring. For targeted drug delivery, a wide range of agents, including antibodies, peptides, and aptamers, have been effectively employed [29,30,31,32,33,34,35,36,37,38,39], while fluorescent dyes and radioactive isotopes have been used for monitoring [40,41]. Notably, antibodies and radioactive isotopes have found extensive clinical applications and have been pivotal in developing drug delivery systems [42,43,44]. Consequently, considerable advancements have been achieved in utilizing these entities for integrated theranostic applications. However, to fully exploit their potential in theranostics, a stable and integrated platform method is imperative to combine and utilize each component effectively.

In this study, we aim to develop theranostic nanoparticles using neutral liposomes that capture doxorubicin as the anticancer drug. The nanoparticle would be conjugated and modularized using micelles labeled with a target-specific antibody and Copper-64 for PET imaging. This approach will enable specific drug delivery to target cancers overexpressing the designated receptor. Using neutral liposomes minimizes non-specific binding, while antibodies maximize target-specific delivery capabilities. Copper-64 utilizes a 1,4,7-triazacyclononane-1,4,7-triacetic acid (NOTA) bifunctional chelator to optimize labeling efficiency and enhance bio-applicability. Through in vitro and in vivo evaluations of these liposomal theranostic nanoparticles, we aim to assess their functionalities and present a potentially applicable platform for future research.

## 2. Results and Discussion

### 2.1. Preparation of ^64^Cu-Immunoliposomes Encapsulating Doxorubicin

^64^Cu-immunoliposomes are achieved through the post-insertion of antibody-micelles and ^64^Cu-NOTA-micelles into liposomes encapsulating doxorubicin (Figure 1). Each module constituting the ^64^Cu-immunoliposomes is manufactured and then purified. First, the chemical conjugation of DSPE-PEG2000-amine with *p*-SCN-*Bn*-NOTA was performed (Appendix A). NOTA is required for copper-64 labeling and is known to have the best labeling efficiency and biocompatibility among the various bifunctional chelators for copper-64 labeling [45]. The conjugate was confirmed through a TLC plate and separated and purified using a silica gel open column. The purified conjugate showed that the molecular weight of DSPE-PEG2000-NOTA was changed from 2859.8 to 3249.1 compared to DSPE-PEG2000-amine through MALDI-TOF/TOF analysis (Appendix A), and about 0.86 NOTAs were bound to one DSPE-PEG2000-amine. The conjugate was confirmed to be *Rf* 0.53 through the TLC plate (Appendix A). DSPE-PEG2000-NOTA produced through this combination secured a labeling rate of more than 97% copper-64 to make ^64^Cu-NOTA-micelles (Appendix A). And micelle composed of mPEG2000-DSPE and mal-PEG2000-DSPE was conjugated with an antibody using maleimide and thiolated cetuximab antibody. The conjugate was separated and purified using a CL-4B column (Appendix A). Lastly, the Dox-liposomes are composed mainly of POPC and cholesterol (Table 1) and encapsulate doxorubicin in a pH-dependent manner.

The encapsulation rate of doxorubicin in the liposome was 98.4%, with a size of 139.7 ± 2.9 nm and a surface charge of −3.1 ± 0.4 (Table 2 and Appendix A). The antibody-conjugated micelles, ^64^Cu-NOTA-micelles, and Dox-liposomes were integrated through the post-insertion method. The process depends on the transition temperature (Tm) of the lipids composing the host liposomes, commonly occurring around 60 °C [46]. However, liposomes based on POPC exhibit a relatively lower Tm (−2 °C) than other neutral liposomes based on phosphatidylcholine [47]. Therefore, POPC/cholesterol liposomes and DSPE-PEG2000 can facilitate an appropriate lipid transition-mediated post-insertion process at 37 °C [48]. This ensures the linker’s or antibodies’ stability while inducing a stable and efficient post-insertion procedure. The integrated ^64^Cu-immunoliposomes were separated and purified by CL-4B column (Appendix A), and the fraction was checked with a gamma counter.

The fifth fraction of ^64^Cu-immunoliposome was isolated and analyzed by iTLC and HPLC. As a result of analyzing the fifth fraction by iTLC, a purification efficiency of more than 98% was confirmed (Appendix A), and HPLC analysis showed the 14.1 min retention time of doxorubicin measured at 495 nm, 14.1 min of antibody measured at 280 nm, and 16.9 min measured with copper-64 radioactivity (Appendix A). The slight increase in retention time measured with copper-64 was due to the physical distance between the radioactivity and absorbance detectors. The physical properties of ^64^Cu-immunoliposomes were analyzed by DLS, showing a size of 147.5 ± 4.0 nm, a surface charge of −2.4 ± 0.6, and a doxorubicin encapsulation yield of 96.1% (Table 2).

### 2.2. In Vitro Analysis of ^64^Cu-Dox-Immunoliposomes

The serum stability of ^64^Cu-immunoliposomes was analyzed for in vivo clinical applications in the future. According to the iTLC results, the integrity of copper-64-labeled immunoliposomes was stably maintained in 50% human serum at 37 °C for 48 h (96.8% stability) (Figure 2A). It shows a sufficiently stable Copper-64 labeling on the surface of the carriers under in vivo conditions. Moreover, ^64^Cu-immunoliposomes showed target-directed cell binding, with 29.8% and 20.5% of the added dose to EGFR-overexpressing A431 cells and intermediate expressing A549 cells, respectively. However, only 1.2% of the added dose was bound to MDA-MB-453 cells not expressing EGFR (Figure 2B). The cell binding analyses with a gamma counter revealed the EGFR-directed cell binding of the liposomal carriers. The EGFR expression level of each cell line was determined using flow cytometry (Appendix A).

Also, the target-dependent intracellular uptake of Dox-immunoliposomes was analyzed with a confocal microscope. A small aliquot of rhodamine-DOPE (0.1 mole%) was included in the liposomes to ensure the tracking of their intracellular uptake. As with the cell binding of the liposomal formulations, it was confirmed that their intracellular uptake intensity was according to the expression rate of EGFR on the cell surface (Figure 2C). At the same time, the cell toxicities of free doxorubicin and immunoliposomal doxorubicin were compared to each other using A549 lung cancer cells. According to the results (Figure 2D), the cytotoxicity of Dox-immunoliposomes (0.40 μM, IC_50_) was slightly higher than free doxorubicin (0.86 μM, IC_50_), supporting the target-dependent intracellular uptake of anti-EGFR immunoliposomes. The in vitro results verified that the anti-EGFR immunoliposomes were more efficiently bound to the cells and transferred into the cytoplasm via the recognition of EGFR on the cell surface, which may enhance the anti-cancer drug efficacy.

### 2.3. In Vivo Therapeutic Efficacy

Tumor xenograft mice were prepared by implanting A549 cancer cells on the mice’s right flanks. The mice were treated with saline, free doxorubicin, empty immunoliposomes, or Dox-immunoliposomes (6 mg doxorubicin/kg) once a week, three times for three weeks, through the tail vein. The treatment began when the tumor size reached about 100 mm^3^. There was no significant difference in tumor growth between the saline-treated and empty immunoliposome-treated groups. However, the Dox-immunoliposome-treated ones began to show a meaningful difference after the second injection and then exhibited a significantly different change in tumor growth after the last injection compared to the free doxorubicin-treated ones (Figure 3A). The mice administered with the suggested dose of Dox-immunoliposomes and free doxorubicin did not show significant body weight loss throughout the treatment period (Figure 3B), implying safe and therapeutic indexes. These results confirmed that the Dox-immunoliposomes could be stably adopted to treat EGFR-expressing tumors in mice.

### 2.4. PET/CT Imaging of ^64^Cu-Immunoliposomes

PET images of ^64^Cu-immunoliposomes were taken at 30 min, 4 h, and 24 h after intravenous administration (*n* = 3) to assess the feasibility of tumor monitoring. The PET images of conventional ^64^Cu-liposomes showed tumor accumulations of 0.11 ± 0.1%ID/g at 30 min, 0.22 ± 0.13%ID/g at 4 h, and 0.01 ± 0.01%ID/g at 24 h. While the PET image of anti-EGFR ^64^Cu-immunoliposome showed more significant tumor accumulation of 0.56 ± 0.3%ID/g at 30 min, 2.2 ± 0.56%ID/g at 4 h, and 0.77 ± 0.31%ID/g at 24 h, compared to the conventional formulation (Figure 4). The most significant (ten times) difference was shown at 4 h post-administration, which then diminished. These differences could be explained as being due to the EGFR-targeting capability of the immunoliposomes in vivo.

### 2.5. Biodistribution of ^64^Cu-Immunoliposomes

The in vivo distribution of liposomes was analyzed at 1 h, 4 h, and 24 h post-administration to A549 tumor-bearing mice (Figure 5 and Appendix A). The blood circulation of the immunoliposomes was maintained for 4 h and then rapidly diminished. Meanwhile, their liver accumulation peaked 4 h later and slowly diminished for 24 h. As previously reported [49,50,51], their half-life in the body is expected to be within 24 h. Nanoparticles are readily opsonized and taken up by macrophages, leading to high accumulation in organs such as the liver and spleen [52,53]. According to the analyses of tumor-to-blood ratios, the tumor accumulation of ^64^Cu-immunoliposomes appeared to maintain stability; 2.9 ± 0.5%ID/g 4 h later and 2.8 ± 0.0%ID/g 24 h later. The tumor-to-blood ratio (T/B) and tumor-to-muscle ratio (T/M) also increased as time elapsed. Comparatively, free cetuximab antibodies circulated longer in blood and consequently showed higher tumor accumulation (Appendix A). The A549 tumor uptake of the copper-64-labeled cetuximab with PCTA linker was 9.2 ± 1.1%ID/g at 48 h post-injection, which appeared to be somehow lower compared to a previous similar report [54]. 

It has been well documented that so-called PEGylation prolongs various nanocarriers, including liposomes [55]. Therefore, the behavior of immunoliposomes in the body was analyzed according to the varied amounts of PEG. The immunoliposomes with more than 5 mole% PEGylation exhibited a slight reduction in particle size and lower doxorubicin encapsulation by about 15% points (Appendix A). While the time-dependent biodistribution patterns in most organs were like 5 mole% PEGylated liposomes, their concentrations in the blood and tumors were elevated by additional PEGylation (Appendix A). The tumor accumulation of the immunoliposomes increased with the enhanced circulation in blood. However, the additional PEGylation had little effect on the T/B and T/M ratios (Figure 5). These perplexing results may require a further systematic in vivo investigation to elucidate how much PEGylation affects blood circulation and the tumor accumulation of anti-EGFR immunoliposomes.

### 2.6. Tumor Tissue Analysis

A549 tumor-bearing mice were injected with Dox-liposomes and Dox-immunoliposomes into the tail vein to analyze their tumor accumulation. Rhodamine-labeled liposomes were injected and, 4 h later, tumor tissues were frozen-sectioned and examined. The microscopic examination showed that the Dox-immunoliposomes were more widely distributed in the tumor tissues than the Dox-liposomes (Figure 6). The distribution of the Dox-immunoliposomes in tumor tissue was further analyzed at various time points after injection (1 h, 4 h, and 24 h). The tile scanning method of confocal microscopy exhibited a higher distribution of Dox-immunoliposomes throughout the tumor as time elapsed. The high mag images also showed more effective intracellular uptake by the individual tumor cells. The immunoliposomes distribution in the tumors widened, and the fluorescence signal became more robust over time (Figure 6B). These results imply that the anti-EGFR immunoliposomes could extravasate into tumor tissues, efficiently spread through interstitial areas, and were then translocated in the target tumor cells.

## 3. Materials and Methods

### 3.1. Reagents

1-palmitoyl-2-oleoyl-sn-glycero-3-phosphocholine (POPC), cholesterol (chol), 1,2-distearoyl-sn-glycero-3-phosphoethanolamine-*N*-[methoxy(polyethylene glycol)-2000] (DSPE-mPEG2000), 1,2-distearoyl-sn-glycero-3-phosphoethanolamine-*N*-[maleimide(polyethylene glycol)-2000] (DSPE-PEG2000-Mal), 1,2-distearoyl-snglycero-3-phosphoethanolamine-*N*-[amino(-polyethyleneglycol)-2000] (DSPE-PEG2000-NH_2_), and 1,2-dioleoyl-snglycero-3-phosphoethanolamine-*N*-(lissamine rhodamine B sulfonyl) (DOPE-rho) were purchased from Avanti Polar Lipids (Alabaster, AL, USA). 2-S-(4-isothiocyanatobenzyl)-1,4,7-triazacyclononane-1,4,7-triacetic acid was purchased (*p*-SCN-*Bn*-NOTA) from Macrocyclics (Dallas, TX, USA). Copper-64 was produced at the Korea Institute of Radiological and Medical Sciences (KIRAMS, Seoul, Republic of Korea) by 50-MeV cyclotron irradiation [56]. Chloroform, methanol, dimethylformamide (DMF), and triethylamine (TEA) were purchased from Sigma-Aldrich (St. Louis, MO, USA). Cetuximab was purchased from Merck KGaA (Darmstadt, Germany). Amicon filters were purchased from Millipore (Billerica, MA, USA).

### 3.2. Cell Lines and Culture

#### 3.2.1. Cell Lines and Cell Culture

A431 (human epidermoid carcinoma), A549 (human lung carcinoma), and MDA-MB-453 (human breast carcinoma) were obtained from the American Type Culture Collection (ATCC). A431, A549, and MDA-MB-453 cells were maintained in DMEM (Welgene Inc., Gyeongsan, Republic of Korea), RPMI-1640 (Welgene), and Leibovitz L-15 (Welgene), respectively. Media were supplemented with 10% fetal bovine serum (Gibco, Grand Island, NY, USA), 100 IU/mL penicillin (Gibco), and 100 μg/mL streptomycin (Gibco). The cells were cultured in a humidified atmosphere of 95% air and 5% CO_2_ at 37 °C, but CO_2_-free in the case of Leibovitz L-15.

#### 3.2.2. Flow Cytometric Analysis

A431, A549, and MDA-MB-453 were analyzed by flow cytometry. In brief, the cells (1 × 10^6^) were incubated with isotype control (Rituximab; Roche, Basel, Switzerland) and cetuximab for 1 h at 4 °C. Then, the cetuximab-treated cells were washed twice with PBS containing 1% BSA (Sigma-Aldrich) and incubated with FITC-conjugated anti-human IgG (Sigma-Aldrich) for 1 h at 4 °C. After washing twice with PBS containing 1% BSA, the treated cells were analyzed using FACSCalibur and CellQuest software (BD Immunocytometry System, San Jose, CA, USA/version 5.1).

### 3.3. Synthesis of DSPE-PEG2000-NOTA

#### 3.3.1. Synthesis Process

DSPE-PEG2000-NH_2_ (M.W.: 2790.486, 50 mg, 17.9 μmol) and *p*-SCN-*Bn*-NOTA (M.W.: 559.9, 30 mg, 53.6 μmol) were dissolved in 1.5 mL of dimethylformamide (DMF) with trimethylamine (TEA) and stirred for 18 h at room temperature. The reaction mixtures were analyzed by thin-layer chromatography (TLC) analysis. One microliter of solution was deposited on a TLC plate 1 cm from the bottom. The solvent (chloroform:methanol:water = 65:25:4) was run to 9 cm from the bottom of the plate, and then the plate was analyzed using UV detection, iodine staining (Sigma-Aldrich), and ninhydrin staining (Sigma-Aldrich) methods. The DSPE-PEG2000-NOTA conjugate was purified using a silica gel open column (solvent; chloroform:methanol:water = 65:25:4) four times. The purified product was then evaporated using an evaporator (BÜCHI Labortechnik AG, Flawil, Switzerland).

#### 3.3.2. MALDI-TOF/TOF MS Analysis

Conjugation of DSPE-PEG2000-NOTA was confirmed by matrix-assisted laser desorption ionization/time of flight (MALDI/TOF) mass spectroscopy. The purified product (1 mg) was added to an ultraflextreme (Bruker Daltonics, Billerica, MA, USA) mass spectrometer using sinapinic acid as a matrix (Bruker Daltonics) in Korea Basic Science Institute (KBSI, Daejeon, Republic of Korea).

### 3.4. Preparation of Antibody-Conjugated Micelles

Cetuximab was thiolated using Traut’s reagent (2-Iminothiolane hydrochloride, Sigma-Aldrich) for 1 h at room temperature in HEPES buffer (25 mM HEPES, 140 mM NaCl, 2 mM EDTA, pH 8.0). Unreacted Traut’s reagent was removed by passing through the PD-10 column (Merck KGaA) with HEPES buffer (25 mM HEPES, 140 mM NaCl, pH 7.4). The thiolated antibodies were conjugated to Mal-PEG2000-DSPE/mPEG2000-DSPE (1:4 molar ratio) micelles at the molar ratio of 0.2:1 (Antibody:Mal-PEG2000-DSPE) by incubation for 16 h at 4 °C with continuous stirring. After the incubation of the antibody with the micelles, the conjugate of cetuximab and lipid was purified using chromatography on sepharose CL-4B columns in HEPES buffer (25 mM HEPES, 140 mM NaCl, pH 7.4). The antibody conjugation yield and purity were measured by Lowry protein assay using a bovine gamma globulin (BGG) standard set (Bio-Rad, Hercules, CA, USA).

### 3.5. Preparation of Copper-64-Labeled Micelles

^64^CuCl_2_ (3.7 MBq, mega becquerel) was added to NOTA-PEG2000-DSPE (7.5 μg) in 107.5 μL of sodium acetate buffer (150 mM, pH 6.5). The reaction mixture was stirred at 37 °C for 1 h with constant shaking using a thermomixer (Eppendorf, Hamburg, Germany). Its radiolabeling yield and radiochemical purity were assessed with instant thin layer chromatography-silica gel (ITLC-SG) strips (Pall Corporation, Port Washington, NY, USA) as a stationary phase and 20 mM citrate buffer pH 5, with 50 mM EDTA as a mobile phase. A half μL of solution was deposited on ITLC-SG strips at 2 cm of the bottom. The solvent was allowed to be 12 cm from the bottom of the strips. Radiolabeled lipids remained at the application point (Ratio-to-front, *Rf* = 0), while free Copper-64 migrated to the solvent front (*Rf* = 1). The strips were then analyzed using a thin-layer chromatography scanner (Bioscan, Poway, CA, USA).

### 3.6. Preparation of ^64^Cu-Dox-Immunoliposomes

#### 3.6.1. Liposome Preparation

POPC (50 mole% or 47.5 mole%), cholesterol (47 mole% or 44.5 mole%), DSPE-mPEG2000 (3 mole% or 8 mole%), and DOPE-rho (0.1 mole%) were dissolved in the chloroform and methanol mixture (2:1, *v*/*v*). The organic solvent was evaporated under a stream of N_2_ gas and the evaporated lipid film was desiccated to ensure the removal of the residual organic solvent by vacuation for 2 h. The dried lipid films were hydrated in an appropriate amount of citrate buffer (150 mM, pH 4.0) and then vigorously mixed by vortexing. After hydration, the dispersion was sonicated for 15 min three times with 30 min intervals. The suspension was subjected to 8 cycles of freezing and thawing under liquid nitrogen and a water bath at 42 °C, then extruded 10 times through a polycarbonate membrane with pore sizes from 200 nm to 80 nm of diameter using an extruder (Avanti polar Lipids) and stored at 4 °C.

#### 3.6.2. Encapsulation of Doxorubicin into Liposomes

The liposomes were hydrated with citrate buffer (150 mM, pH 4.0) and passed through the PD-10 column equilibrated with the HEPES buffer (25 mM HEPES, 140 mM NaCl, pH 7.4) to replace the outside liposomal solution. Doxorubicin in normal saline was added to the liposomal solution (1:3, weight ratio of doxorubicin and lipid), which was then incubated at 60 °C for 10 min with 100 rpm in a water bath. The incubation process at 60 °C was necessary for the rapid and complete entrapment of doxorubicin vesicles inside. The separation of liposomal doxorubicin from free doxorubicin was performed by PD-10 column equilibrated with HEPES buffer (25 mM HEPES, 140 mM NaCl, pH 7.4). The doxorubicin concentration was determined by absorbance at 495 nm after lysis of the Dox-liposome with Triton-X 100 (0.5%, *v*/*v*, Sigma-Aldrich).

#### 3.6.3. Post Insertion of Antibody-Micelles and ^64^Cu-Micelles to Dox-Liposomes

For the insertion of anti-EGFR antibody-lipid conjugates and ^64^Cu-NOTA-lipid conjugates into the outer membrane of liposomes, the micelles containing antibody-PEG2000-DSPE or ^64^Cu-NOTA-PEG2000-DSPE lipid were incubated with preformed liposomes encapsulating doxorubicin for 4 h at 37 °C with continuous stirring. Uninserted free antibody-PEG2000-DSPE or ^64^Cu-NOTA-PEG2000-DSPE were removed from the ^64^Cu-immunoliposomes encapsulating doxorubicin by gel filtration chromatography through sepharose CL-4B columns in HEPES buffer (25 mM HEPES, 140 mM NaCl, pH 7.4). The radioactivity of each fraction was counted in a gamma counter (WIZARD 1480, Perkin-Elmer, Waltham, MA, USA). Radiochemical purity was also analyzed by size-exclusion high-performance liquid chromatography (SE-HPLC) using a MAbPac SEC-1 column (Thermo Scientific). The mobile phase consisted of 0.3 M NaCl in 50 mM sodium phosphate buffer pH 6.8 and the column was eluted at a flow rate of 0.5 mL/min. The retention time of ^64^Cu-immunoliposomes was determined at 280 nm of UV absorbance. The radioactivity of ^64^Cu-immunoliposomes was determined by a radioactivity detector (Raytest, Straubenhardt, Germany).

#### 3.6.4. Characterization of ^64^Cu-Dox-Immunoliposomes

The sizes and ζ-potentials of liposomes, immunoliposomes, and their doxorubicin-containing formulations were measured using a particle analyzer (Malvern Instruments Ltd., Malvern, UK). Each sample (500 μL of 1 mg/mL liposomes) in HEPES buffer (25 mM HEPES, 140 mM NaCl, pH 7.4) was loaded to cuvettes and measured 3 times by a particle analyzer.

### 3.7. In Vitro Serum Stability

^64^Cu-immunoliposomes were added to an equal volume of human serum (total volume of 100 μL) and incubated at 37 °C for 48 h. At the indicated time points (0, 3, 9, 12, 24, and 48 h post-incubation), the radioactivity of ^64^Cu-immunoliposomes or dissociated Copper-64 was analyzed by ITLC-SG analysis. The stability of liposomes was calculated by comparing the radioactivity of ^64^Cu-immunoliposomes at each time point and the initial radioactivity at 0 h (*n* = 3).

### 3.8. In Vitro Cell Binding Assay

To evaluate the cell binding activity of ^64^Cu-immunoliposomes to various cancer cell lines (A431, A549, MDA-MB-453), an in vitro cell binding assay was performed. Cell binding studies with ^64^Cu-immunoliposomes (400 pmole of phospholipid) were carried out at 4 °C for 1 h using A431, A549, and MDA-MB-453 cells (1 × 10^6^ cells in 5 mL tube, a total of 300 μL, triplicate). Nonspecific binding was determined in the presence of 66 μg/mL of cetuximab under the same experimental conditions. After incubation, the samples were washed twice in cold PBS containing 0.1% BSA. The radioactivity of each sample was counted in a γ-counter. Cell bound radioactivity (%) was calculated by (Cell bound radioactivity − Nonspecific radioactivity)/Total radioactivity × 100.

### 3.9. In Vitro Cellular Uptake

A431, A549, and MDA-MB-453 cells were seeded on coverslips in 6-well plates (2.5 × 10^5^ cells/well) and cultured for 24 h. The prepared rhodamine-labeled liposomes containing doxorubicin were added to the cells, which were then incubated at 4 °C for 30 min. After removing the medium, the treated cells were washed twice with cold PBS (pH 7.4). The cells were stained with one drop of 4′,6-diamidino-2-phenylindole (DAPI) solution (Vector Lab., Burlingame, CA, USA) for 30 min in a dark place and mounted on slides. The slides were observed using a laser confocal scanning microscope (LSM 510, Zeiss, Heidenheim, Germany).

### 3.10. In Vitro Cytotoxicity

A549 cells (5 × 10^4^ cells/well) were seeded in 12-well plates and incubated for 18 h. The cells were treated with ^64^Cu-immunoliposomes encapsulating varied concentrations of doxorubicin in culture medium (50, 100, 500, 1000, 5000, and 10,000 ng/mL) with 10% FBS. Control cells were treated with media alone or doxorubicin alone. After 48 h of cell incubation at 37 °C, the cytotoxicity of free doxorubicin and ^64^Cu-immunoliposomes encapsulating doxorubicin was evaluated by determining cell viability by the trypan blue exclusion assay. Data were expressed as a percentage of control proliferation using the number of living cells incubated with media alone as a control and all experiments were performed in triplicate.

### 3.11. Preparation of Tumor Xenograft Mouse Model

All animal experiments were conducted under a protocol approved by Wonju College of Medicine Institutional Animal Care and Use Committee at Yonsei University (IACUC, #YWC-160211-1). Female BALB/c-nude mice (Orient Bio, Seongnam, Republic of Korea), aged 5 weeks, were used in all experiments. A549 cells of 1 × 10^7^ in 0.1 mL of PBS were injected subcutaneously into the right flank of each mouse.

### 3.12. Antitumor Efficacy

When the tumor volume reached ~100 mm^3^, A549 tumor-bearing mice were randomly divided into four groups (*n* = 4~5 per group). Each group was treated with saline, doxorubicin, liposomes containing doxorubicin, immunoliposomes containing doxorubicin, and control liposomes without doxorubicin. The amount of drug injected in each mouse was 6 mg/kg equivalent to doxorubicin. Tumor volume was calculated by (longitudinal diameter × height^2^)/2. Tumor volume and body weight were measured three times a week.

### 3.13. Micro-PET Imaging

To evaluate the tumor targeting of ^64^Cu-immunoliposomes, PET and CT (computed tomography) imaging was performed in A549 tumor-bearing mice. ^64^Cu-immunoliposomes or ^64^Cu-liposomes (3.15 ± 0.27 MBq) were intravenously administered into the mice and static scans were acquired for 10 min at 30 min, 4 h, and 24 h (1 h for 24 h) after injection using a small animal PET scanner (Inveon, Siemens, Munich, Germany). Quantitative data were expressed as a percentage of injected dose per gram of tissue (%ID/g). Image visualization was performed using Inveon Research Workplace (Siemens).

### 3.14. Tissue Distribution Analysis

#### 3.14.1. Bio-Distribution Study of ^64^Cu-PCTA-Cetuximab

Nude mice were subcutaneously injected with the A549 lung cancer cell line (5 × 10^6^ cells) in the right hind flank. When tumor xenografts were fully established and had reached volumes of around 100 mm^3^, the mice were intravenously administered with a 3.7 MBq (50 μg) of ^64^Cu-PCTA-cetuximab and sacrificed at 48 h. The blood, various tissues, and A549 tumors were excised and weighed. The radioactivity was measured using a gamma counter (WIZARD 1480, Perkin-Elmer) applying a decay correction. Counts were compared with those of standards, and the data were expressed as the percentage of injected radioactivity dose per gram of tissue (%ID/g).

#### 3.14.2. Bio-Distribution Study of ^64^Cu-Immunoliposomes

Nude mice were subcutaneously injected with the A549 lung cancer cell line (5 × 10^6^ cells) in the right hind flank. When tumor xenografts were fully established and had reached volumes of around 100 mm^3^, 3 MBq of PEG 5 mole% ^64^Cu-immunoliposomes or PEG 10 mole% ^64^Cu-immunoliposomes were intravenously administered into each mouse (*n* = 3). At different times (1 h, 4 h, and 24 h) after intravenous injection, mice were sacrificed by CO_2_ asphyxiation. Blood samples were collected through cardiac puncture. Organs of interest were removed and weighed with radioactivity measured by a gamma counter (WIZARD 1480, Perkin-Elmer). The results were expressed as the percentage of injected dose per gram of tissue (%ID/g).

### 3.15. Confocal Microscopic Analysis of Liposome-Administered Tumor Tissue

The tumors were explanted 4 h or various time points (1 h, 4 h, and 24 h) after Dox-immunoliposomes or Dox-liposomes (360 μg of liposomes) intravenous administration in A549 tumor-bearing mice and preserved in 4% paraformaldehyde for 3 h at 4 °C. Then, the tumor tissues were transferred to the 30% sucrose PBS until the tissues were soaked at 4 °C. The tissues transfer to the 30% sucrose with tissue freezing compound (1:1 *v*/*v*, Tissue Tek OCT compound, Sakura Finetek, Torrance, CA, USA) for 30 min at 4 °C and the tissues were preserved in OCT compound at −80 °C. Five-micrometer-thick slides were processed for confocal imaging. The tumor nuclei were mounted with a mounting medium containing DAPI (Vectashield Mounting Medium with DAPI, Vector Laboratories, Burlingame, CA, USA). Fluorescence microscopy was performed to visualize the rhodamine-liposome (yellow/Cy3 channel), doxorubicin (red/FITC channel), and DAPI (blue channel) using a Zeiss LSM510 confocal microscopy system (Carl Zeiss GmbH, Aalen, Germany) equipped with krypton–argon and ultraviolet lasers, and the images were acquired using LSM version 5 software (Carl Zeiss).

### 3.16. Statistical Analysis

Quantitative data were represented as mean ± standard deviation (S.D.) and statistical analysis was performed by one-way and two-way ANOVA or Student’s *t*-test using Prism 6 (GraphPad Software, La Jolla, CA, USA). *p* values of < 0.05 were considered statistically significant.

## 4. Conclusions

In this study, we successfully developed and demonstrated a theranostic liposomal system simultaneously capable of treating and imaging EGFR-expressing lung cancer. To achieve this, the radioactive isotope copper-64 was chelated to NOTA-PEG-lipid micelles (monitoring module). At the same time, anti-EGFR antibodies were also conjugated to PEG-lipid micelles (targeting module). Both modules were inserted into POPC/cholesterol-based neutral liposomes encapsulating doxorubicin (therapeutic vector module). 

The incorporation of NOTA- and antibody-micelles into the liposomes was achieved by the post-insertion method, resulting in the final product of ^64^Cu-Dox-immunoliposomes. The liposomes displayed a size of approximately 150 nm, a slightly negative charge, and 96% doxorubicin encapsulation efficiency. Copper-64 and antibodies were stably coupled to the liposomal surface under in vitro- and in vivo-mimicking conditions. In addition, the ^64^Cu-Dox-immunoliposomes exhibited EGFR-specific binding to target cancer cells (A431 and A549) with the least off-target binding (MDA-MB-453). Through in vivo studies with mice carrying target A549 tumors, the anti-EGFR immunoliposomes could be more effectively accumulated in tumors, providing effective tumor imaging and treatment.

The ^64^Cu-Dox-immunoliposomes modularly prepared in this study would be a versatile platform applicable to various scenarios involving theranostic nanoparticles. The modular nature of the preparation method allows for easy application to different target systems and diseases. Therefore, this platform has great potential to offer diverse possibilities for future targeted therapies and diagnostic imaging applications for advanced precision medicine.

## Figures and Tables

**Figure 1 ijms-25-01813-f001:**
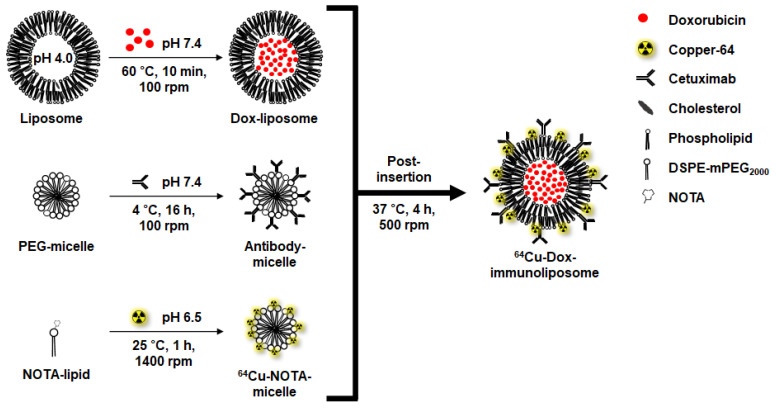
Schematic illustration of ^64^Cu-immunoliposomes encapsulating doxorubicin. Doxorubicin encapsulating liposomes, antibody-conjugated micelles, and Copper-64-labeled NOTA-micelles were combined by post-insertion at 37 °C for 4 h in a 500 rpm agitator.

**Figure 2 ijms-25-01813-f002:**
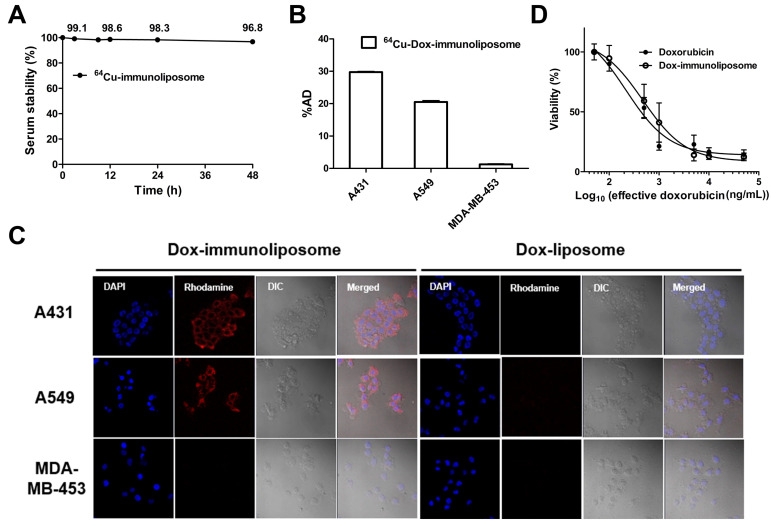
In vitro analysis of ^64^Cu-Dox-immunoliposomes. (**A**) In vitro copper-64 labeling stabilities of ^64^Cu-Dox-immunoliposomes at specific times after incubation in 50% human serum at 37 °C (mean ± SD, *n* = 3) were analyzed with iTLC. (**B**) Quantitative cellular binding of the liposomes to A431, A549, and MDA-MB-453 was evaluated using a gamma counter, and (**C**) the cellular uptake of the Dox-immunoliposomes and Dox-liposomes containing rhodamine was observed using confocal microscopy (×400). (**D**) In vitro cytotoxicities of free doxorubicin and Dox-immunoliposomes in A549 cells were confirmed by MTT assay, and IC_50_ values of them were 0.40 µM and 0.86 µM. %AD: added dose.

**Figure 3 ijms-25-01813-f003:**
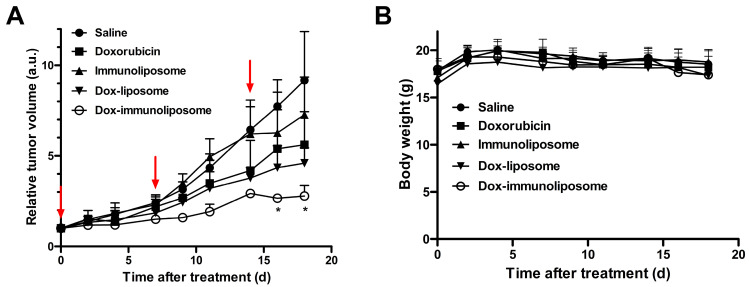
Therapeutic efficacy of anti-EGFR immunoliposomes encapsulating doxorubicin in A549 tumor xenograft model. (**A**) Tumor growth inhibition in A549 tumor xenograft-bearing nude mice after different treatments (*n* = 5). The free doxorubicin and the all formulations of liposomal doxorubicin were intravenously administered at a total dose of 6 mg doxorubicin/kg on the indicated days post tumor implantation (red arrows) and the empty immunoliposomes were injected at the same liposome dose of the Dox-immunoliposomes (18 mg of liposome/kg). * *p* < 0.05, vs. Doxorubicin. (**B**) The body weights of the treated mice were measured. Points, mean tumor volumes; bars, SE. *p* values were based on a multivariate (rank) test for the indicated treatment groups.

**Figure 4 ijms-25-01813-f004:**
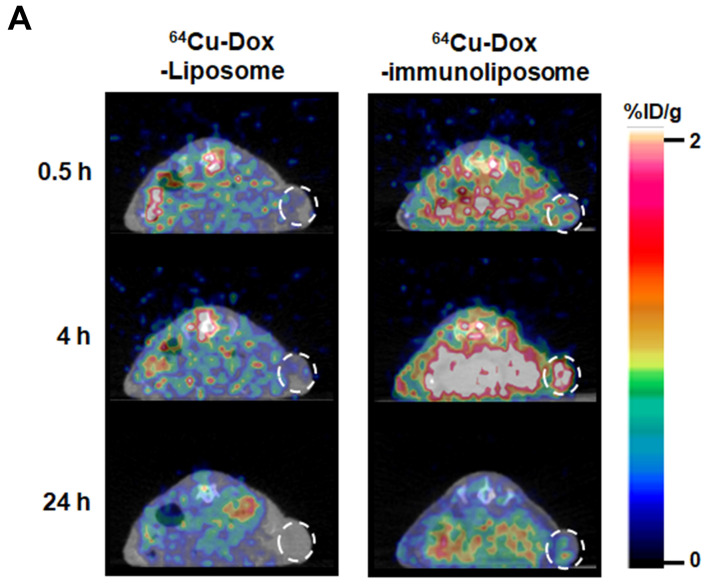
PET/CT images of A549 tumor-bearing mice after intravenous injection of ^64^Cu-Dox-liposomes or ^64^Cu-Dox-immunoliposomes. (**A**) A549 tumor-bearing mice were administrated with 3.15 ± 0.27 MBq of ^64^Cu-Dox-liposomes or ^64^Cu-Dox-immunoliposomes and serially imaged at each time point. White dotted circles indicate the tumors. (**B**) The tumor uptakes of the various liposomes were quantified by small animal PET scans (*n* = 3, * *p* < 0.001) and represented as %ID/g.

**Figure 5 ijms-25-01813-f005:**
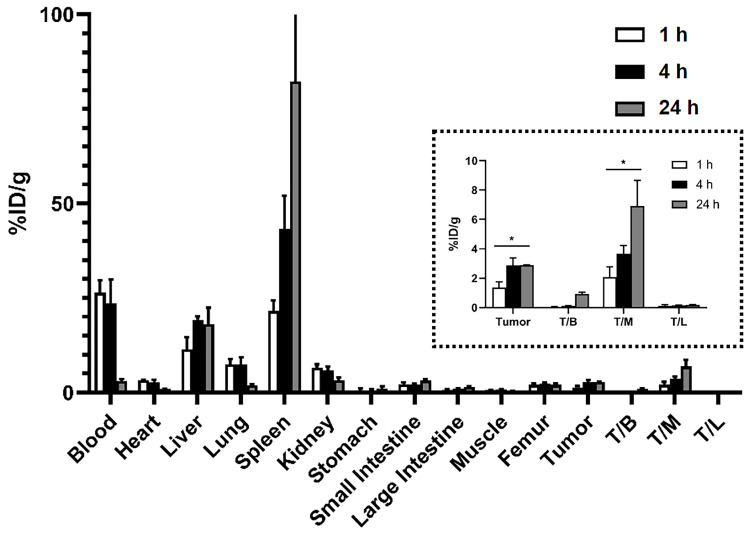
Biodistribution of ^64^Cu-Dox-immunoliposomes in various tissues after intravenous injection in A549 tumor-bearing mice. Mice were sacrificed at 1 h, 4 h, and 24 h after the liposome administration. Tissues (blood, heart, liver, lung, spleen, kidney, stomach, intestine, muscle, femur, tumor, tumor-to-blood, tumor-to-muscle, and tumor-to-liver) were extracted for gamma counting and weight measurement (*n* = 3). Tumor-to-organ ratios of ^64^Cu-Dox-immunoliposomes at 1 h, 4 h, and 24 h post intravenous injection were indicated in the dotted box. The uptake values are expressed as the percentage of injected dose per gram (%ID/g ± SD, * *p* < 0.05).

**Figure 6 ijms-25-01813-f006:**
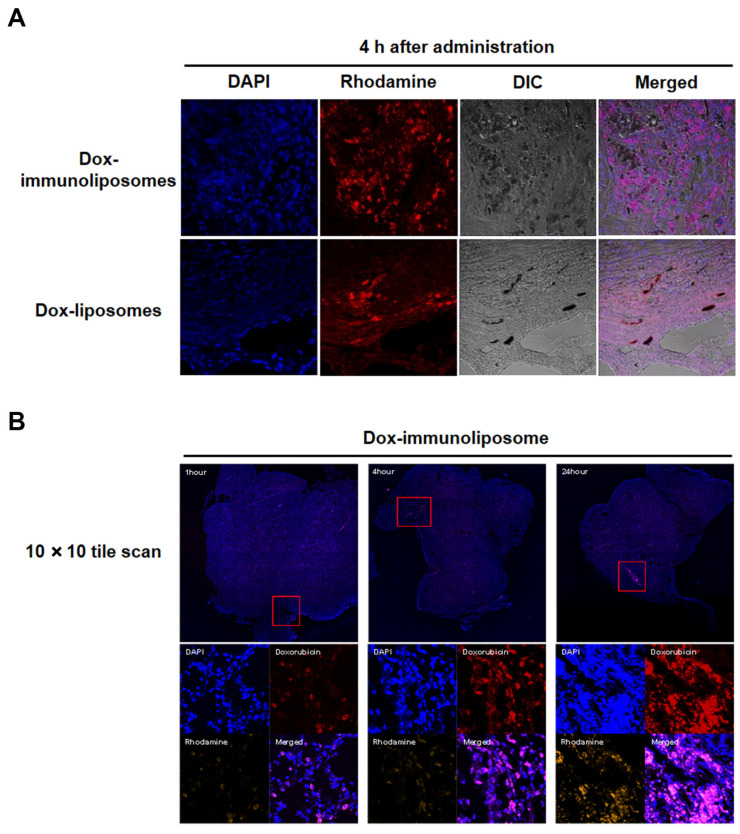
Confocal microscopy images of tumor tissues after intravenously administered Dox-immunoliposomes and Dox-liposomes. (**A**) Dox-immunoliposomes and Dox-liposomes were visualized in the tumor tissue at 4 h post intravenously administered in A549 tumor-bearing mice with the rhodamine (red, rhodamine-labeled liposomes; ×40 original magnification) channel. (**B**) Dox-immunoliposomes were injected via the tail vein and the tumor tissues were cryo-sectioned 1 h, 4 h, or 24 h later. The Dox-immunoliposomes were visualized in the tumor tissues with the rhodamine (yellow, rhodamine-labeled liposomes) and Alexa-fluor 488 (red, doxorubicin) channels. The images were visualized with 10 × 10 tile scanning and ×40 original magnification scanning. DAPI staining for nuclei is shown in all pictures.

**Table 1 ijms-25-01813-t001:** Components of liposomes and PEG-micelles.

Nanoparticles	Lipid Components	Ratio (Mole%)
Liposomes	POPC	50
	Cholesterol	47
	DSPE-mPEG2000	2.9
	Rho-DOPE	0.1
PEG-micelles	DSPE-mPEG2000	80
	DSPE-PEG2000-Mal	20

**Table 2 ijms-25-01813-t002:** Physicochemical properties of prepared liposomes.

Nanoparticles	Size (nm) *	Zeta-Potential (mV) *	Polydispersity Index	Doxorubicin Encapsulation Efficiency (%)
Liposomes (3%PEG)	154.8 ± 2.4 **	−3.6 ± 0.4 **	0.07	-
Immunoliposomes (5%PEG)	154.3 ± 2.5	−4.2 ± 1.4	0.14	-
Dox-liposomes (3%PEG)	139.7 ± 2.9	−3.1 ± 0.4	0.08	98.4
Dox-immunoliposomes (5%PEG)	147.5 ± 4.0	−2.4 ± 0.6	0.06	96.1

Notes: * The particle size and zeta-potentials were measured 3 times using a zeta sizer. ** The particle size (nm); average particle size ± S.D. Zeta potential (mV); average zeta-potential ± S.D.

## Data Availability

The datasets and materials used and/or analyzed during the current study are available from the corresponding author upon reasonable request.

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
