# Peer review of "Development of PET Radioisotope Copper-64-Labeled Theranostic Immunoliposomes for EGFR Overexpressing Cancer-Targeted Therapy and Imaging"

_ijms, 2024, doi:10.3390/ijms25031813_

Round 1
Reviewer 1 Report
Comments and Suggestions for Authors
Dear Authors,
This manuscript is interesting. Development of PET radioisotope copper-64-labeled theranostic immunoliposomes for EGFR overexpressing cancer- targeted therapy and imaging by Jeong et al describes the liposomal nanoparticles as theranostic agents capable of simultaneous therapy and imaging of target cancer cells. The authors selected copper-64 (64Cu), with a clinical l half-life (t1/2 = 12.7 h) and decay properties, as the radioisotope for molecular PET imaging. An anti-epidermal growth factor receptor (anti-EGFR) antibody was used to achieve target-specific delivery. the chemotherapeutic agent doxorubicin (Dox) was encapsulated within the liposomes using a pH-gradient method.
Positive comments:
1. The conjugates of 64Cu-labeled and anti-EGFR antibody-conjugated micelles were inserted into the doxorubicin-encapsulating liposomes via post-insertion procedure (64Cu-Dox-immunoliposomes) and this will target mainly cancer cells.
2. The size and zeta-potential of the liposomes and analyzed target-specific cell binding and cytotoxicity in EGFR-positive cell lines.
3. The authors analyzed the specific therapeutic effect and PET imaging of the 64Cu-Dox-immunoliposomes with the A549 xenograft mice model.
4. They performed In vivo therapeutic experiments on the mice models demonstrated that the doxorubicin-containing 64Cu-immunoliposomes effectively inhibited tumor growth.
5. Moreover, the 64Cu-immunoliposomes provided superior in vivo PET images of the tumors compared to the untargeted liposomes.
Comments on the Quality of English Language
Very minor proof reading required.
Reviewer 2 Report
Comments and Suggestions for Authors
The authors presented the paper "Development of PET radioisotope copper-64-labeled theranostic immunoliposomes for EGFR overexpressing cancer- targeted therapy and imaging" Thank you for so interesting work. However, I have some questions and comments.
1) Section 2.1 and Figure 1. How you can be sure that you have antibodies on the outer part of liposome? It is an important point. Why it can't be inside? Have you calculated someway how many antibodies in your liposomes (per liposome)?
2) Table 2 Please, present pictures of number, volume, and intensity modes in SI. This size is a number mode? Intensity is more relevant to show the homogeneity of the sample. Why you have so similar size and charge (zeta potential) for all systems? Moreover, such small zeta-potential usually indicates not good stability of dispersion. How you can comment it?
Moreover, in Table 2 Dox-immunoliposomes (5%PEG) is the resulted (final) liposomes with copper?
3) I see 50% human serum stability experiment. However, have you analyzed protein binding on liposomes surface, which may highly influence their further bio distribution? Moreover, it looks very strange that human serum albumin can't bind copper efficient from your complexes? Have you studied copper binding constant for your complex and interaction with albumin?
4) Cell cytotoxicity. Please, mention in Figure 2D caption that it is A549 cells. Moreover, you have similar to DOX results. How you explain it? Why you liposomes are not more effective than free DOX? It’s very easy to get the impression that they don’t work. According to Figure 2D, DOX works better and your better results for IC50 calculation can be because of the high error and not enough experimental points.
5) Figure 5. Please, explain in the paper text why you have so much amount in liver and spleen. Is it possible really to see such small amount in tumor? Why your system not works?
Comments on the Quality of English Language
Minor editing of English language required
Round 2
Reviewer 2 Report
Comments and Suggestions for Authors
Thank you for the revised paper.
Comments on the Quality of English Language
Minor editing of English language required